# The influence of social support, physical activity, general psychological distress, and demographic characteristics on self-reported health status among women in Iran

Badriyeh Karami[1], Shahab Rezaeian[2,3], Ebrahim Shakiba[4,5], Amirhossien Naghibzadeh[6], Abbas Mohammad Karimi Mazhin[7], Masoumeh Malek[8], Hadi Darvishigilan[9]*

1 Health Care Services Management. Behavioral Diseases Research Center, Health Institute, Kermanshah University of Medical Sciences, Kermanshah, Iran, 2 Epidemiology. Infectious Diseases Research Center, Health Institute, Kermanshah University of Medical Sciences, Kermanshah, Iran, 3 Epidemiology, Department of Epidemiology, School of Health, Kermanshah University of Medical Sciences, Kermanshah, Iran, 4 Clinical Biochemistry, Department of Biochemistry, School of Medicine, Kermanshah University of Medical Sciences, Kermanshah, Iran, 5 Clinical Biochemistry, Behavioral Diseases Research Center, Kermanshah University of Medical Sciences, Kermanshah, Iran, 6 Nursing. Student Research Committee, Kermanshah University of Medical Sciences, Kermanshah, Iran, 7 Student in Health in Emergency and Disaster, Social Welfare and Rehabilitation Sciences University, Tehran, Iran, 8 Nursing, Department of Anesthesia , Tehran School of Allied Medical Sciences, Tehran University of Medical Sciences, Tehran, Iran, 9 Health Education and Promotion, Health and Environment Research Center, Ilam University of Medical Sciences, Ilam, Iran

* hadidarvishi2000@yahoo.com

## Abstract

### Background

Considering the impact of social support on women's mental health and the importance of their mental health in improving the health of the society, this study aimed to investigate the influence of social support, physical activity, general psychological distress, and demographic characteristics on self-reported health status among women.

### Methods

This cross-sectional study was conducted on 350 women aged 18–75 in 2024 in Iran. The Vaux social support, international physical activities, Depression- Anxiety- Stress Scale (DASS-21) questionnaires were administered. The Cluster random sampling method was used. The data were analyzed using STATA version 18. The significance level for examining the hypotheses was $p \geq 0.05$.

### Results

According to the results, there was a significant positive relationship between age, place of residence, history of chronic disease, evaluation of financial status and self-rated health in both crude and adjusted models. In addition, the positive significant

**Data availability statement:** All relevant data are within the paper and its Supporting Information file.

**Funding:** The author(s) received no specific funding for this work.

**Competing interests:** The authors have declared that no competing interests exist.

relationship was observed between general psychological distress, and poor self-rated health in adjusted model. So that, the odds of poor self- rated health in women who achieved higher depression, anxiety, stress scale score was 1.04 times higher than others.

## Conclusion

Considering the rising rates of psychological problems, particularly in recent years among women, as well as the influence of women's health on societal well-being, it is imperative for policymakers in women's health to pinpoint the underlying causes of mental disorders in women. They should take proactive measures to mitigate these causes, thereby decreasing the prevalence of anxiety, depression, and stress disorders through the implementation of effective interventions.

## Introduction

Health is a fundamental human right and a key indicator of justice within society [1]. The right to health encompasses the ability to lead a healthy and productive life of good quality, characterized by an acceptable lifespan and freedom from illness and disability, with the responsibility for this right resting on governments [2]. Consequently, a primary objective of the health system in every nation is to ensure, sustain, and enhance the health and quality of life for all societal members [3]. In this context, the physical and mental well-being of women holds significant importance due to their professional and social contributions to society [4]. The health of women directly influences the health of their family members, particularly their children; thus, neglecting women's health can adversely affect future generations [5].

In a 2023 study conducted by Armandpisheh et al., it was found that the rates of stress, depression, and anxiety were greater in women than in men [6]. This observation aligns with earlier research carried out in Iran, which indicated that women are more susceptible to psychiatric disorders [7]. Furthermore, such disorders are also found to be more common in women than in men in various other nations [5]. Factors such as socioeconomic disadvantages, sex hormones, cultural disparities, and violence contribute to the increased prevalence of psychiatric disorders among women [8].

Self-rated health (SRH) serves as a dependable and straightforward metric for assessing general health [9] and is an effective predictor of mortality and individual health status [10]. Research has demonstrated that individuals' health perceptions largely correspond with evaluations made by healthcare professionals [11]. Numerous factors have been identified as influencing SRH, including demographic, socioeconomic, behavioral, psychological, and disease-related elements [12]. Significant socioeconomic disparities in SRH have been noted, with those of lower socioeconomic status reporting worse SRH outcomes [13].

SRH, also referred to as self-perceived health, is a singular health assessment tool where individuals evaluate their current health condition on a scale of four or five

points, ranging from excellent to poor [14]. Prior research indicates that SRH serves as a reliable predictor of mortality associated with various illnesses [15]. Furthermore, numerous scholars have sought to investigate the factors linked to SRH, discovering a strong correlation between SRH and both morbidity [16] and disability [17]. Previous studies have revealed that individuals' self-evaluation of their health across multiple physical, psychological, and social aspects is statistically significant in relation to social support; those with favorable overall social support tend to report better self-assessments of their health [18,19]. Additionally, physical activity is considered a crucial factor influencing health status, encompassing cardiovascular health, skeletal strength, and psychological wellness [20].

Health behaviors encompass actions or inactions that have a direct or indirect impact on health [21]. These behaviors include habits associated with a healthy lifestyle, such as sleep and exercise [22]. Additionally, certain studies indicate that economic or social factors play a significant role as determinants [23]. Consequently, gaining insight into the factors linked to self-rated health can assist professionals in prioritizing interventions for health promotion and disease prevention [24]. SRH serves as a subjective assessment of health status, often referred to as "perceived" or "subjective" health. This concept has been extensively examined in survey research [25]. Nevertheless, the majority of studies concerning SRH have concentrated on specific age groups, gender groups, or patient populations [26,27].

Currently, in light of evolving trends impacting health and disease, it is crucial to comprehend the elements influencing women's health. Based on the results of our research, limited studies have examined self-rated health status among women in past years in Iran [28,29], but there has been no study that investigates the association between social support, physical activity, and psychological problems related to self- rated health among women in Iran. Therefore, given the significance of disease prevention among women, this study sought to assess the status of SRH and its correlation with social support, physical activity, and general psychological distress, considering various socio-demographic factors in women.

## Methods

### Study design and setting

This cross-sectional study was carried out in 2024 involving women aged 18–75 residing in Kermanshah city. Following the acquisition of the ethical code, and considering the socio-economic disparities across different areas of Kermanshah city, data collection was executed separately for the eight regions of the city. The method employed for sampling and selecting neighborhoods was cluster random sampling. Utilizing a random number table, two neighborhoods were randomly chosen from each of the eight regions of Kermanshah city, resulting in a total of 16 neighborhoods.

After obtaining the ethics code from the ethical committee of KUMS (ethics code: IR.KUMS.REC.1403.078), the data collection process began. Data collection was conducted from May 30 to October 15, 2024. Data were gathered in each neighborhood by visiting households and completing questionnaires by women aged 18–75. Households were included in the study based on the sample size calculated for each region in relation to the population. The inclusion criteria for participation in the study encompassed individuals aged 18–75 years, possessing adequate memory and capability to respond to the questionnaire items, while the exclusion criteria comprised a documented history of mental illnesses such as depression as diagnosed by a physician, a history of acute illnesses including cancer, stroke, cardiovascular diseases, etc., significant physical impairments that restricted physical activity, being pregnant, and a lack of willingness to participate in the study after receiving a comprehensive explanation of the research and its objectives, along with assurances regarding the confidentiality of the information provided. For participants which was unable to complete the questionnaire for any reason, trained interviewers were employed and asked the questions and then recorded their answers, confidentially.

### Participants

To determine the sample size, the correlation coefficient of 0.24, which reflects the relationship between social support and SRH as reported in the study by Movahed et al., [30] was employed. Utilizing the sample size formula, it was

projected that 290 women would be necessary. Considering a 20% non-response rate for the study, the final estimated sample size was determined to be at least 348 individuals. In the Sahebi et al.'s study [31], the eight districts of Kermanshah city were classified into three categories: poor districts [2,3,9], medium districts [1,5,6], and good districts [4,8] based on the livability index, which encompasses three dimensions: socio-cultural, economic, and environmental. Consequently, participants were chosen utilizing the cluster random sampling technique. Each district comprises multiple neighborhoods that differ greatly in terms of socio economic status. Therefore, according to the research team's decision, two neighborhoods were randomly chosen from each district (totally 16 neighborhoods. The requirements for participating in the study included women aged 18–75 years, willing to participate in the study, not having sensory perception disorders or mental retardation, and the ability to speak and understand Persian or Kurdish.

## Variables

The primary instrument for data collection in this research was a questionnaire that comprised five distinct sections. *The initial section* gathered demographic details about the participants, which included relevant past behaviors such as their history of professional physical activity, involvement in sports classes, and the duration of time spent watching television and playing computer games. Additionally, personal factors were assessed, including age, gender, marital status, body mass index, educational level, occupation, place of residence, region of residence, income, previous medical history, and a self-comparison with others regarding education level, occupation, and income level.

*The second section* featured the International Physical Activity Questionnaire (IPAQ), designed to evaluate physical activity over the preceding 7 days through a self-reported format consisting of 6 questions. This section categorized individuals into three groups: inactive, minimally active, and highly active. The questionnaire further classified activities into three types: intense (questions 1 and 2), moderate (questions 3 and 4), and walking (questions 5 and 6). For intense activities, a coefficient of 8 was assigned, while moderate activities received a coefficient of 4, and moderate walking was assigned a coefficient of 3.3. After establishing these coefficients, the daily time spent on each activity type and the number of days per week dedicated to that activity were calculated and multiplied [32,33]. Ultimately, the resulting figures were input into STATA in a similar manner.

*The third section* was the Depression, Anxiety, and Stress Scale (DASS-21) for measuring general psychological distress and symptoms related to depression, anxiety, and stress. This scale was created by Lovibond in 1995, and its validity and reliability have been previously assessed by Moradipanah et al., in Iran [34]. The questionnaire consists of 21 items, which include 8 items pertaining to depression (D), 7 items concerning anxiety (A), and 6 items related to stress (S). In this assessment, individuals were instructed to indicate their moods over the past week based on the statements provided in the questionnaire, categorized into four groups. The response options were organized into four categories: (0; not at all), (1; little), (2; much), and (3; very much). The correlation of this scale with the Beck Depression Inventory (BDI) and the Eysenck Anxiety Inventory was found to be significant. The alpha values in a sample of 400 were reported as 70% for depression, 66% for anxiety, and 76% for stress, with the correlation of the depression subscale with the Beck Depression Inventory (BDI) being 0.70, the anxiety subscale with the Eysenck Anxiety Inventory at 0.75, and the stress subscale with the Perceived Stress Inventory at 0.49 [35].

*The fourth section* pertains to the Vaux Social Support Questionnaire. This questionnaire was created by Vaux et al., [36] utilizing Cobb's definition of social support. Its validity was assessed in Iran during the research conducted by Ebrahimi Ghavam et al., [37] involving 100 and 200 students. The reliability of the test within the student sample was determined to be 0.90 for the overall scale, 0.7 for the student sample, and 0.81 in a retest conducted six weeks later. In the research carried out by Karimi et al., [38] and colleagues, the alpha coefficient calculated for this questionnaire was found to be 0.74. This questionnaire is composed of three dimensions and includes 23 questions. Each dimension concerning social support from friends and family was evaluated with 8 questions, while the dimension related to other sources of

social support was assessed with 7 questions. The scoring for the questions is as follows: strongly disagree [1], disagree [2], no opinion [3], agree [4], and strongly agree [5]. A higher score signifies greater social support.

*The fifth section* was a survey question created by the World Health Organization (WHO) for measuring Self-rated health (SRH) [39]. This survey question serving as a tool for predicting mortality rates in populations both with and without cardiovascular disease [39,40]. In numerous countries, surveys that prompt participants to evaluate their overall health on a five-point scale (ranging from excellent to poor) have gained popularity as a health indicator [41,42]. Participants were inquired, "how do you feel about your health?", and their health status was classified according to the five-point Likert scale: "very good", "good", "fair", "poor", or "very poor". Utilizing this scale, SRH is categorized into poor SRH and good SRH [39].

## Statistical analysis

Descriptive statistics, including frequency, percentage, mean, and standard deviation, were calculated to summarize participants' demographic and study variables. The dependent variable, self-rated health (SRH), originally measured on a 5-point Likert scale (from very poor to very good), was dichotomized into two categories: poor SRH (very poor/poor/fair) and good SRH (good/very good). This dichotomization was performed to facilitate binary logistic regression analysis and is consistent with prior studies using SRH as a global health indicator. To examine the association between demographic characteristics, social support, physical activity, and psychological problems with SRH, binary logistic regression analysis was performed. Initially, univariable (crude) logistic regression models were fitted to assess the unadjusted associations between each independent variable and SRH. Variables with a p-value < 0.20 in the crude analysis were entered into the multivariable (adjusted) model to control for potential confounding effects. Before running the final model, multicollinearity among independent variables was examined using the variance inflation factor (VIF) and tolerance statistics, with VIF values < 2.5 indicating acceptable collinearity. The overall model fit was evaluated using the Hosmer–Lemeshow goodness-of-fit test and Nagelkerke $R^2$ to assess the explanatory power of the model. Results of the logistic regression are reported as odds ratios (ORs) with 95% confidence intervals (CIs). A two-tailed p-value < 0.05 was considered statistically significant. All analyses were performed using STATA version 18.

## Results

### Participants

According to the results of Table 1, most of the participants were under 30 years (46%), married (55.71%), urban (91.42%). 67.14% had an academic degree. In addition, 41.42% received health information by social networks. 67.14% had history of chronic disease. 41.42% didn't do any social activities, typically. 69.14% of participants were evaluated their self-rated health status as good.

### Main results

As shown in Table 2, there were a significant positive relationship between age, place of residence, history of chronic disease, evaluation of your financial status and self-rated health in both crude and adjusted models. So that, the odds of poor SRH in older women was 1.28 times higher than others (OR=1.28, CI: [1.04, 1.57], p < 0.001) and odds of poor SRH in those with higher education status was 1.27 times higher than others women (OR=1.27, CI: [0.89, 1.82], p < 0.001). In addition, the odds of poor SRH in women who live in rural region was 1.27 times higher than women who live in urban region (OR=1.27, CI: [1.06, 6.83], p < 0.03) and the odds of poor SRH in women who evaluate your financial status at very good status was 0.11 times more than others (OR=0.11, CI: [0.46, 0.94], p < 0.02).

Furthermore, the positive significant relationship was observed between physical activity, DASS and poor SRH in adjusted model. So that, the odds of poor SRH in women who had more physical activity was 0.59 times higher than others. Also, the odds of poor SRH in women who achieved higher DASS score was 1.04 times higher than others.

**Table 1. Demographic characteristics of the respondents by self- rated health status.**

| Demographic characteristics | | Self- rated health status | |
|---|---|---|---|
| | | Good, N (%) | Poor, N (%) |
| Self-rated health status | | 242 (69.14) | 108 (30.86) |
| Age | <24 | 71(29.34) | 8 (7.41) |
| | 25-29 | 63 (26.03) | 19 (17.59) |
| | 30-34 | 22 (9.09) | 10 (9.26) |
| | 35-39 | 30 (12.4) | 25 (23.15) |
| | 40-45 | 27 (11.16) | 21(19.44) |
| | 46-51 | 13 (5.37) | 5 (4.63) |
| | >52 | 16 (6.61) | 20 (18.52) |
| Education | Preliminary | 10 (4.13) | 16 (14.81) |
| | Secondary (guidance) | 11(4.55) | 7 (6.48) |
| | High school | 48 (19.83) | 23 (21.3) |
| | Academic | 173 (71.49) | 62 (57.41) |
| Marital status | Single | 128 (52.89) | 27 (25) |
| | Married | 114 (47.11) | 81(75) |
| Job status | Retired | 16 (6.61) | 15 (13.89) |
| | Unemployed | 8 (3.31) | 7 (6.48) |
| | Employed | 63 (26.03) | 27 (25) |
| | Other | 155 (64.04) | 69 (35.96) |
| Place of residence | Urban | 227 (93.8) | 93 (86.11) |
| | Rural | 15 (6.2) | 15 (13.89) |
| Home ownership | Private | 185 (76.45) | 76 (70.37) |
| | Rental | 44 (18.18) | 24 (22.22) |
| | Other | 13 (5.37) | 8 (7.41) |
| Child numbers | 0 | 158 (65.29) | 36 (33.33) |
| | 1 | 33 (13.64) | 29 (26.85) |
| | 2 | 33 (13.64) | 22 (20.37) |
| | 3 | 18 (7.44) | 21(19.44) |
| Supplementary insurance | Yes | 85 (35.12) | 52 (48.15) |
| | No | 157 (64.88) | 56 (51.85) |
| Source of received health information | Public Media | 45 (18.6) | 29 (26.85) |
| | Physician | 80 (33.06) | 36 (33.33) |
| | Social Networks | 106 (43.8) | 39 (36.11) |
| | Other | 11 (4.55) | 4 (3.7) |
| History of chronic disease | Yes | 52 (21.49) | 63 (58.33) |
| | No | 190 (78.51) | 45 (41.67) |
| Evaluation of your financial status | Very good | 13 (5.37) | 3 (2.78) |
| | Good | 61 (25.21) | 6 (5.56) |
| | Not good, not bad | 115 (47.52) | 60 (55.56) |
| | Poor | 35 (14.46) | 30 (27.78) |
| | Very poor | 18 (7.44) | 9 (8.33) |
| People who support you | Family | 171 (70.66) | 82 (75.93) |
| | Closed relatives | 13 (5.37) | 8 (7.41) |
| | Friend | 58 (23.97) | 18 (16.67) |

*(Continued)*

**Table 1.** (Continued)

| Demographic characteristics | | Self- rated health status | |
|---|---|---|---|
| | | **Good, N (%)** | **Poor, N (%)** |
| Social activities you do typically | Charitable Activities | 12 (4.96) | 6 (5.56) |
| | Non-profit Activities | 29 (11.98) | 14 (12.96) |
| | Religious Activities | 15 (6.2) | 14 (12.96) |
| | Public Sports Activities | 17 (7.02) | 3 (2.78) |
| | No Activities | 139 (57.44) | 56 (51.85) |
| | Mixed activities | 30 (12.4) | 15 (13.89) |

## Discussion

This study aimed to investigate the role of social support, physical activity, psychological disorders, and demographic characteristics in SRH among women. Most participants rated their health as "good" (69.14%). The findings indicated several results that illuminate the significance of social support, depression, anxiety, stress, and demographic factors in influencing self-rated health. Following the adjustment for possible confounding variables, the research discovered that the depression, anxiety, stress score, and demographic factors such as age, place of residence, history of chronic disease, and favorable financial status were linked to poor SRH.

In crude model **SRH** were significantly associated with **social support**. This is consistent with the results of a study by Matud et al., which states that women and men who had higher social support had better self-rated health [43]. In addition, in this study, the relationship between general **psychological distress** and **SRH** was statistically significant and in the research conducted by Zhang et al., [44] psychosomatic ailments emerged as the predominant factors in establishing a rating framework for self-rated health status. The results from the research conducted by Barkhordari-Sharifabad et al., indicated that social support is inversely related to symptoms of anxiety and depression [45]. Gutiérrez-Sánchez et al., [46] and Tadayon et al., [47] demonstrated in their study that receiving sufficient social support correlates with an improved quality of life and reduced levels of depression when compared to those who receive less social support. Social support can be regarded as a significant factor in mitigating the risk of developing mental health disorders. It is noted that social support is inversely related to symptoms of depression and anxiety in individuals [45], and those who have social support tend to experience greater efficacy along with reduced anxiety and depression [48].

Particularly, in circumstances where social support is minimal, symptoms of premenstrual syndrome can become exacerbated [49]. This suggests that women lacking social support may exert less effort in addressing their issues, resulting in diminished success in their endeavors. In essence, the absence of understanding and social support from their surroundings adversely affects women's performance and diminishes their sense of worth and self-esteem. This chain of events can lead to depression and the experience of stress and pressure [50], ultimately influencing their SRH.

Furthermore, findings from additional studies indicated that an increase in the variety of social support received by women correlates with a rise in their engagement in leisure-time physical activities. This outcome aligns with the majority of domestic [51–53] and international [54–56] research, while contrasting with study conducted by Soto et al., [57]. The discrepancies observed may stem from regional differences where the studies were conducted, as well as variations in cultural contexts and participant characteristics. In the current research, the majority of participants possessed a university degree. Previous research indicates that higher education can enhance problem-solving capabilities and skills, improve individuals' understanding and analytical abilities, and boost self-confidence levels. Individuals with higher education experience increased social freedom and garner more respect and support [58]. Conversely, educated women tend to participate more in recreational sports activities than their less-educated counterparts. Although individuals with varying educational backgrounds encounter obstacles, the participation rate

**Table 2. Logistic regression analysis the relationship between characteristics of the respondents, social support, and physical activity, depression, anxiety, stress, and poor self-rated health (SRH).**

| Variables | | Poor Self-Rated Health (SRH) | | | | | | | |
|---|---|---|---|---|---|---|---|---|---|
| | | Crude model | | | | Adjusted model | | | |
| | | OR | SE | P > \|z\| | [95% CI] | OR | SE | P > \|z\| | [95% CI] |
| Age | | 1.38 | 0.08 | 0.00 | 1.23, 1.56 | 1.28 | 0.13 | 0.01 | 1.04, 1.57 |
| Education | | 0.64 | 0.08 | 0.00 | 0.5, 0.82 | 1.27 | 0.23 | 0.17 | 0.89, 1.82 |
| Marital status | Single | Ref. | – | – | – | – | – | – | – |
| | Married | 3.36 | 0.86 | 0.00 | 2.03, 5.57 | 1.81 | 0.7 | 0.12 | 0.85, 3.89 |
| Place of residence | Urban | Ref. | – | – | – | – | – | – | – |
| | Rural | 2.44 | 0.94 | 0.02 | 1.14, 5.19 | 2.69 | 1.27 | 0.03 | 1.06, 6.83 |
| Home ownership | Private | Ref. | – | – | – | – | – | – | – |
| | Rental | 1.32 | 0.38 | 0.32 | 0.75, 2.33 | | | | |
| | Other | 1.49 | 0.7 | 0.38 | 0.59, 3.76 | | | | |
| Child numbers | | 1.71 | 0.18 | 0.00 | 1.38, 2.12 | 1.04 | 0.19 | 0.81 | 0.72, 1.51 |
| Supplementary insurance | No | Ref. | – | – | – | – | – | – | – |
| | Yes | 1.71 | 0.4 | 0.02 | 1.08, 2.71 | 1.54 | 0.45 | 0.13 | 0.87, 2.74 |
| Source of received health information | Public Media | Ref. | – | – | – | – | – | – | – |
| | Physician | 1.43 | 0.44 | 0.24 | 0.77, 2.63 | – | – | – | – |
| | Social Networks | 0.81 | 0.22 | 0.46 | 0.47, 1.4 | – | – | – | – |
| | Other | 0.8 | 0.49 | 0.73 | 0.24, 2.71 | – | – | – | – |
| History of chronic disease | No | Ref. | – | – | – | – | – | – | – |
| | Yes | 5.11 | 1.27 | 0.00 | 3.13, 8.35 | 2.95 | 0.89 | 0.00 | 1.63, 5.36 |
| Evaluation of your financial status | | 0.62 | 0.08 | 0.00 | 0.48, 0.8 | 0.66 | 0.11 | 0.02 | 0.46, 0.94 |
| People who support you | Family | Ref. | – | – | – | – | – | – | – |
| | Closed relatives | 1.28 | 0.6 | 0.59 | 0.51, 3.21 | 0.54 | 0.31 | 0.3 | 0.17, 1.71 |
| | Friends | 0.64 | 0.19 | 0.14 | 0.35, 1.16 | 0.6 | 0.21 | 0.16 | 0.29, 1.23 |
| Social activities you do typically | No Activities | Ref. | – | – | – | – | – | – | – |
| | Charitable Activities | 1.24 | 0.65 | 0.68 | 0.44, 3.46 | – | – | – | – |
| | Non-profit Activities | 1.19 | 0.43 | 0.61 | 0.58, 2.43 | – | – | – | – |
| | Religious Activities | 2.31 | 0.93 | 0.03 | 1.04, 5.11 | – | – | – | – |
| | Public Sports Activities | 0.43 | 0.28 | 0.2 | 0.12, 1.55 | – | – | – | – |
| | Mixed activities | 1.24 | 0.43 | 0.54 | 0.62, 2.48 | – | – | – | – |
| Social Support | | 1.04 | 0.00 | 0.00 | 1.02, 1.06 | 1.01 | 0.01 | 0.18 | 0.99, 1.04 |
| Physical Activity | | 0.93 | 0.21 | 0.77 | 0.59, 1.47 | 0.59 | 0.17 | 0.07 | 0.33, 1.06 |
| Depression, Anxiety, Stress | | 1.05 | 0.01 | 0.00 | 1.03, 1.07 | 1.04 | 0.01 | 0.001 | 1.02, 1.07 |

in sports among those with education levels exceeding a bachelor's degree is significantly higher than that of others [59]. This increased propensity for physical activity can positively influence individuals' physical and mental well-being and, as a result, their self-perception of health.

According to the results, SRH were significantly associated with **age**. Which is in line with the results of the study by Movahed Majd et al., [60]. In such a way that in the aforementioned study, the individual's self-rated health decreases with increasing age. Generally, health tends to decline as we grow older. This occurs due to physiological changes in the body and a gradual decrease in organ function over time [61]. The velocity and magnitude of these transformations are not uniform across various individuals, and additional determinants such as lifestyle choices, genetic predispositions, and environmental influences also contribute to the overall health of each person [62]. In our study, most participants were

under 40 years of age, they received social support from family and friends, did not report a history of chronic disease, so they reported good self-rated health.

In addition, there was a significant relationship between **place of residence** and **SRH**. The presence of this noteworthy correlation between social support and self-rated health has been substantiated in the research conducted by Movahed Majd et al., [60] and as social support intensifies, the individual's self-perceived health assessment has concurrently risen.

**SRH** were significantly associated with **good financial status**. In the results of a study conducted among women by Azmand et al., [28] the adequacy of income for living expenses was stated as an influential factor on SRH. In additional research, health was also found to have a substantial correlation with the financial status of individuals [63]. Insufficient income subjects individuals to a greater array of stressors, including inadequate living conditions and economic strain, thereby increasing their vulnerability to mental health disorders [64].

## Limitations

One of the limitations of this cross-sectional study is the inability to establish causality. This type of research merely indicates the correlation between variables rather than a definitive cause-and-effect relationship, making it impossible to assert with certainty that one factor led to a change in another. Additionally, another limitation of this study arises from the fact that the variables were gathered via a questionnaire, and the data were obtained through self-reported measures, which introduces the potential for response bias. We acknowledge that the wide interval may introduce heterogeneity. Stratifying participants into narrower age groups could provide more precise insights into age-specific determinants of self-rated health. We will consider this recommendation in future studies and have noted it as a limitation of the current work.

## Conclusion

Overall, the results of this research suggest that women's self- rated health is notably affected by a mix of demographic, socioeconomic, and health-related factors. In addition, psychological problems remained a significant determinant in the adjusted model, such that higher levels of depression, anxiety, and stress were associated with increased odds of poor self-rated health. These findings highlight the essential importance of mental health in conjunction with socioeconomic and clinical elements in influencing individuals' views on their overall health, stressing the necessity for a holistic approach to both physical and psychological well-being. Considering the rising rates of anxiety, depression, and stress-related disorders, particularly in recent years among women, as well as the influence of women's health on societal well-being, it is imperative for policymakers in women's health to pinpoint the underlying causes of mental disorders in women. They should take proactive measures to mitigate these causes, thereby decreasing the prevalence of anxiety, depression, and stress disorders through the implementation of effective interventions. Conducting impactful educational workshops focused on women's mental health, enhancing and providing access to sports facilities in community settings, offering social support from pertinent organizations, and establishing active non-governmental organizations dedicated to women's issues are among the effective strategies that can be pursued.

## Supporting information

**S1 File. Results.**

(RAR)

## Author contributions

**Conceptualization:** Badriyeh Karami, Hadi Darvishigilan.

**Data curation:** Shahab Rezaeian, Amirhossien Naghibzadeh, Masoumeh Malek.

**Formal analysis:** Shahab Rezaeian.

**Writing – original draft:** Badriyeh Karami, Ebrahim Shakiba, Abbas Mohammad Karimi Mazhin.

**Writing – review & editing:** Badriyeh Karami, Ebrahim Shakiba, Abbas Mohammad Karimi Mazhin.

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
