## [Editor Report · Decision Letter 0]

9 Oct 2025

Dear Dr. Darvishigilan,

Thank you for submitting your manuscript to PLOS ONE. After careful consideration, we feel that it has merit but does not fully meet PLOS ONE’s publication criteria as it currently stands. Therefore, we invite you to submit a revised version of the manuscript that addresses the points raised during the review process.

**ACADEMIC EDITOR:**

After reviewing the editorial, I found that the study topic is relevant; however, several important issues need to be addressed before further consideration.Foremost, although an ethics code (IR.KUMS.REC.1403.078) is provided, there is no IRCT registration number. Please clarify it.Next, the statistical analysis needs to be revised and clarified. The logistic regression model lacks information on variable selection, control for confounders, assumption testing (normality, multiple collinearity), and goodness-of-fit assessment. Confidence intervals for several odds ratios are missing, and the SRH duality needs to be justified.3rd, please confirm obviously that the study population and dataset are completely independent of the population used in your previously published article in Scientific Reports (2025; DOI: 10.1038/s41598-024-79835-9)Lastly, the manuscript requires significant improvements in language and formatting, including the correction of grammatical errors, consistent verb tense usage, and proper reference formatting.Please review the manuscript in its entirety and provide a detailed and point-by-point response. The revised version will be reevaluated for compliance with PLOS ONE's guidelines to be processed by reviewers.

We look forward to receiving your revised manuscript.

Kind regards,

Zahra Lorigooini

Academic Editor

PLOS ONE
---

## [Author Response · Author response to Decision Letter 1]

15 Oct 2025

Dear editorial board of Journal of Plos One

Thanks for providing the comments of the respectful reviewer to us. We tried to revise the manuscript, titled “The Influence of Social Support, Physical Activity, psychological problems, and Demographic Characteristics on Self-Reported Health Status among Women” based on the comments and respond it in the following table. Revisions has been shown as highlight in the manuscript. Hope the revisions are satisfactory now. However, we welcome any further constructive comments if required.

Dr. Hadi Darvishigilan

Corresponding author

Comments Response

1. Foremost, although an ethics code (IR.KUMS.REC.1403.078) is provided, there is no IRCT registration number. Please clarify it. Thank you for your comments. This study is not a clinical trial study, so it does not have an IRCT code.

This study is the result of a research project conducted with code 4030146 from Kermanshah University of Medical Sciences.

2. The statistical analysis needs to be revised and clarified. The logistic regression model lacks information on variable selection, control for confounders, assumption testing (normality, multiple collinearity), and goodness-of-fit assessment. Confidence intervals for several odds ratios are missing, and the SRH duality needs to be justified. We appreciate your valuable comments. The section on Statistical analysis has been thoroughly revised to provide greater clarity and methodological transparency.

3. Please confirm obviously that the study population and dataset are completely independent of the population used in your previously published article in Scientific Reports (2025; DOI: 10.1038/s41598-024-79835-9) The mentioned study (DOI: 10.1038/s41598-024-79835-9) is the result of a research project with the code 4020003, which was conducted in early 2023 with the participation of 495 people aged 18 to 75 (men and women). This is while the submitted manuscript is the result of another research project with the code 4030146, which was conducted in 2024. The participants were 350 women aged 18 to 75, and the data from this study is completely independent of the study published in the journal Scientific Reports.

4. Lastly, the manuscript requires significant improvements in language and formatting, including the correction of grammatical errors, consistent verb tense usage, and proper reference formatting. The manuscript was improved in language and formatting.

---

## [Decision Letter · Decision Letter 1]

5 Dec 2025

Dear Dr. Darvishigilan,

Thank you for submitting your manuscript to PLOS ONE. After careful consideration, we feel that it has merit but does not fully meet PLOS ONE’s publication criteria as it currently stands. Therefore, we invite you to submit a revised version of the manuscript that addresses the points raised during the review process.

We look forward to receiving your revised manuscript.

Kind regards,

Zahra Lorigooini

Academic Editor

PLOS One

**Journal Requirements:**

Reviewers' comments:

Reviewer's Responses to Questions

**Comments to the Author**

Reviewer #1: All comments have been addressed

Reviewer #2: (No Response)

Reviewer #3: (No Response)

2. Is the manuscript technically sound, and do the data support the conclusions?

Reviewer #1: Yes

Reviewer #2: Yes

Reviewer #3: Partly

3. Has the statistical analysis been performed appropriately and rigorously?

Reviewer #1: Yes

Reviewer #2: Yes

Reviewer #3: I Don't Know

4. Have the authors made all data underlying the findings in their manuscript fully available?

Reviewer #1: Yes

Reviewer #2: Yes

Reviewer #3: No

5. Is the manuscript presented in an intelligible fashion and written in standard English?

Reviewer #1: Yes

Reviewer #2: Yes

Reviewer #3: Yes

Reviewer #1:

To the Editor

PLOS ONE

Dear Editor,

Thank you very much for inviting me to read the paper titled, " The Influence of Social Support, Physical Activity, psychological problems, and Demographic Characteristics on Self-Reported Health Status among Women".

Overall, as a reader of the paper, I felt that the paper is very well written and have important content for professionals in the relevant fields.

Congratulations to the authors for their wonderful work.

Reviewer #2: General Comments:

- The topic is very interesting and relevant to contemporary women’s health issues. The authors aimed to assess how social support, physical activity, mental health, and sociodemographic characteristics among women affect their self-perceived health status. For this goal, the authors used appropriate tools.

- Self-reported health is a crucial data source for healthcare providers, researchers, and decision-makers.

- We suggest specifying the context of the study in the title and abstract (in Iran).

- We recommend using the STROBE checklist for observational studies to report the study.

Specific comments:

Introduction

Overall, the introduction is well written and clear. The authors described the relevance of the topic. However, the existence of 5 concepts (4 exposures and 1 outcome) in the study may make things hard to assimilate. Therefore, please specify why the authors chose to assess these four exposures in relation to self-reported health.

Line 99-101: The idea expression is complex to understand. Please avoid using too many linking adverbs consecutively (additionally, consequently, etc.)

Line 109: Provide a reference to research conducted that reveals that studies concerning the factors associated with SRH in Iran are quite scarce.

Methods

I don't agree with the methods’ section structure. We recommend using the STROBE checklist for observational studies to report on the study, especially the methods section.

Line 116: The research should have one design, descriptive or analytic. In this case, the study is an analytic cross-sectional. The authors could simply write “cross-sectional study”.

Line 129: The authors should report the minimal sample size is 348, and report 350 in the results section.

Line 145: women included in this were aged 18 to 75 years. I can see a huge difference in age in this population, and I wonder if a 75-year-old woman can recall her daily life habits for physical activity, or if she can do physical activity. I think that this age interval is so large that it can affect the study result because of the difference in age characteristics.

-Are pregnant women included in the study? Pregnant women have a specific need for social support and adaptive physical activity. This should be specified in the inclusion and exclusion criteria.

Line 152: Using two different data collection strategies (questionnaire for participants and interview for others) in the study creates information bias. How did the authors manage this?

Line 174: DASS is a measurement tool for mental health status. “Mental health” is more appropriate than “psychological disorders.”

Line 211-213: I do not agree with the authors to perform binary logistic regression should be performed because OR is applicable for case-control studies. However, linear logistic regression is correct and suitable for this study. We recommend performing a multivariable linear regression analysis or at least providing a reliable argumentation regarding the choice of logistic regression.

Results

As mentioned above, I think that the age interval (18 to 75 years) is too large, which can affect the study results due to the difference in age characteristics (developmental patterns are different in each age category). The authors would rather present the data according to age groups (young and adult women (18 to 40 years, middle adulthood women 41 to 65 years, and old women aged more than 65 years).

Discussion

Discussion is missing limitations of the study and its implications for practice

Reviewer #3: References used in the manuscript are outdated. I recommend updating the literature with more recent studies(3-5 years)

**Do you want your identity to be public for this peer review?** For information about this choice, including consent withdrawal, please see our Privacy Policy

Reviewer #1: No

Reviewer #2: **Yes:** Dr Maha Dardouri

Reviewer #3: **Yes:** mobin ebrahimain

---

## [Author Response · Author response to Decision Letter 2]

23 Dec 2025

Dear editorial board of Journal of Plos One

Thanks for providing the comments of the respectful reviewer to us. We tried to revise the manuscript, titled “The Influence of Social Support, Physical Activity, General Psychological Distress, and Demographic Characteristics on Self-Reported Health Status among Women in Iran” based on the comments and respond it in the following table. Revisions has been shown as highlight in the manuscript. Hope the revisions are satisfactory now. However, we welcome any further constructive comments if required.

Dr. Hadi Darvishigilan

Corresponding author

Comments Response

Reviewer 1

1. Overall, as a reader of the paper, I felt that the paper is very well written and have important content for professionals in the relevant fields.

Congratulations to the authors for their wonderful work. Many thanks for your attention and times.

Reviewer 2

1. We suggest specifying the context of the study in the title and abstract (in Iran). Many thanks for your comments. It was conducted.

Page 1, line 2 and Page 2, line 41.

Introduction

1. Overall, the introduction is well written and clear. The authors described the relevance of the topic. However, the existence of 5 concepts (4 exposures and 1 outcome) in the study may make things hard to assimilate. Therefore, please specify why the authors chose to assess these four exposures in relation to self-reported health. Many thanks for your comments. Explanations for the reason for choosing four exposures in relation to self-reported health were added to the end of the introduction.

Page 4, lines 106-109 and Page 5, lines 110-112.

2. Line 99-101: The idea expression is complex to understand. Please avoid using too many linking adverbs consecutively (additionally, consequently, etc.) Many thanks for your comments. It was revised. Page 4, lines 96-101.

3. Line 109: Provide a reference to research conducted that reveals that studies concerning the factors associated with SRH in Iran are quite scarce. Many thanks for your comments. It was conducted.

Page 4, lines 106-107.

Methods

1. I don't agree with the methods’ section structure. We recommend using the STROBE checklist for observational studies to report on the study, especially the methods section. Many thanks for your comments. The method's section structure was revised based on the STROBE checklist.

2. Line 116: The research should have one design, descriptive or analytic. In this case, the study is an analytic cross-sectional. The authors could simply write “cross-sectional study”. Many thanks for your comments. It was revised. Page 1, line 2. Page 2 , line 41.

3. Line 129: The authors should report the minimal sample size is 348, and report 350 in the results section. Many thanks for your comments. It was revised. Page 1, line 41 and Page 5, line 116.

4. Line 145: women included in this were aged 18 to 75 years. I can see a huge difference in age in this population, and I wonder if a 75-year-old woman can recall her daily life habits for physical activity, or if she can do physical activity. I think that this age interval is so large that it can affect the study result because of the difference in age characteristics. The fact that the reviewer noted the good variety of age (18-75 years) of participants and the possibility of the differences between the memories of daily habits and physical activity among older women is insightful. The wide age bracket was chosen deliberately to get a rosy image of self-rated health in women within the community. Age was a covariate in crude and adjusted models of logistic regression, and its strong correlation with self-rated health was controlled. We acknowledge that this broad range can bring about heterogeneity and thus the future research can be stratified by organizing participants into smaller age groups in order to offer age specific information.

This limitation was added to the limitations section.

Page 13, lines 326-330.

5. Are pregnant women included in the study? Pregnant women have a specific need for social support and adaptive physical activity. This should be specified in the inclusion and exclusion criteria. Many thanks for your comments. Pregnant women were not included in this study and were an exclusion criterion. Being pregnant was added to the exclusion criteria in the Methods section.

Page 5, line 131.

6. Line 152: Using two different data collection strategies (questionnaire for participants and interview for others) in the study creates information bias. How did the authors manage this? Many thanks for your comments. The data collection method was not very different, and only in cases where participants were unable to complete the questionnaire for any reason, questions were asked by interviewers and then their responses were recorded confidentially. Hence, there is no possibility of bias.

More explanation added to the method section.

Page 6, lines 134-135.

7. Line 174: DASS is a measurement tool for mental health status. “Mental health” is more appropriate than “psychological disorders.” Many thanks for your comments. Based on the original version of the Depression Anxiety Stress Scales– 21 (DASS-21), this scale is 21-item self-report measure designed to assess the severity of general psychological distress and symptoms related to depression, anxiety, and stress in adults older adolescents (17 years +). So the phrase of “psychological disorders” in the title and other sections was edited.

Page 1, lines 1 and page 2, line 39 and page 7, line 173.

8. Line 211-213: I do not agree with the authors to perform binary logistic regression should be performed because OR is applicable for case-control studies. However, linear logistic regression is correct and suitable for this study. We recommend performing a multivariable linear regression analysis or at least providing a reliable argumentation regarding the choice of logistic regression. We sincerely thank the reviewer for this valuable comment regarding the choice of regression model. In our study, the dependent variable (self-rated health) was dichotomized into two categories (poor vs good SRH). For binary outcomes, logistic regression is widely recommended as the appropriate statistical method, since it estimates odds ratios to quantify the association between predictors and the likelihood of poor SRH. This approach is consistent with established statistical guidelines (Hosmer & Lemeshow, Applied Logistic Regression, 3rd edition).

Page 19, line 217.

Results

1. As mentioned above, I think that the age interval (18 to 75 years) is too large, which can affect the study results due to the difference in age characteristics (developmental patterns are different in each age category). The authors would rather present the data according to age groups (young and adult women (18 to 40 years, middle adulthood women 41 to 65 years, and old women aged more than 65 years). We thank the reviewer for raising this point again. As noted in our response to the previous comment regarding the age variable, all analyses were performed accordingly to control for the confounding effect of age. We added this point as a limitation in our study. Please kindly refer to our earlier response for details.

Page 13, lines 326-330.

Discussion

1. Discussion is missing limitations of the study and its implications for practice. Many thanks for your comments. The limitation section was added to the end of discussion.

Page 13, lines 326-330.

Reviewer 3

1. References used in the manuscript are outdated. I recommend updating the literature with more recent studies (3-5 years). Many thanks for your comments. All references were revised and updated as much as possible.

---

## [Decision Letter · Decision Letter 2]

5 Jan 2026

Dear Dr. Darvishigilan,

Thank you for submitting your manuscript to PLOS ONE. After careful consideration, we feel that it has merit but does not fully meet PLOS ONE’s publication criteria as it currently stands. Therefore, we invite you to submit a revised version of the manuscript that addresses the points raised during the review process.

We look forward to receiving your revised manuscript.

Kind regards,

Zahra Lorigooini

Academic Editor

PLOS One

Journal Requirements:

Reviewers' comments:

Reviewer's Responses to Questions

**Comments to the Author**

Reviewer #2: All comments have been addressed

Reviewer #3: All comments have been addressed

2. Is the manuscript technically sound, and do the data support the conclusions?

Reviewer #2: Yes

Reviewer #3: Partly

3. Has the statistical analysis been performed appropriately and rigorously?

Reviewer #2: Yes

Reviewer #3: I Don't Know

4. Have the authors made all data underlying the findings in their manuscript fully available?

Reviewer #2: Yes

Reviewer #3: No

5. Is the manuscript presented in an intelligible fashion and written in standard English?

Reviewer #2: Yes

Reviewer #3: Yes

Reviewer #2: We would like to thank the authors for addressing all comments. The manuscript now is more relevant and suitable for publication.

Reviewer #3: As previously noted, it is recommended that the references in this section be updated to primarily include sources published within the last 3–5 years, in order to better reflect the current state of knowledge.

**Do you want your identity to be public for this peer review?** For information about this choice, including consent withdrawal, please see our Privacy Policy

Reviewer #2: **Yes:** Dr Maha Dardouri

Reviewer #3: **Yes:** Mobin Ebrahimian

---

## [Author Response · Author response to Decision Letter 3]

7 Jan 2026

Dear editorial board of Journal of Plos One

Thanks for providing the comments of the respectful reviewer to us. We tried to revise the manuscript, titled “The Influence of Social Support, Physical Activity, General Psychological Distress, and Demographic Characteristics on Self-Reported Health Status among Women in Iran” based on the comments and respond it in the following table. Revisions has been shown as highlight in the manuscript. Hope the revisions are satisfactory now. However, we welcome any further constructive comments if required.

Dr. Hadi Darvishigilan

Corresponding author

Comments to the Author Response

1. If the authors have adequately addressed your comments raised in a previous round of review and you feel that this manuscript is now acceptable for publication, you may indicate that here to bypass the “Comments to the Author” section, enter your conflict of interest statement in the “Confidential to Editor” section, and submit your "Accept" recommendation.

Reviewer 1:

All comments have been addressed Many thanks for your attention and time.

Reviewer 2:

All comments have been addressed Many thanks for your attention and time.

2. Is the manuscript technically sound, and do the data support the conclusions?

Reviewer 1:

Yes. Many thanks for your attention and time.

Reviewer 2:

Partly. Many thanks for your comment. The conclusion section was revised.

3. Has the statistical analysis been performed appropriately and rigorously?

Reviewer 1:

Yes. Many thanks for your attention and time.

Reviewer 2:

I Don't Know. Many thanks for your comment. Considering that Dr. Shahab Rezaian (PhD in Epidemiology) has served as a statistical consultant and one of the study authors in various studies and has sufficient skills and expertise in this field, all statistical analysis of the study were performed and rechecked by him. Also, the other authors have sufficient skills and expertise in interpreting and analyzing the results, so there is no concern about the accuracy and precision of the statistical analysis.

4. Have the authors made all data underlying the findings in their manuscript fully available?

Reviewer 1:

Yes. Many thanks for your attention and time.

Reviewer 2:

No. Many thanks for your comment. As stated in the manuscript and in the submission system, all underlying study findings are available in the manuscript, and in the event of any ambiguity, the raw data of the study that has been analyzed and analyzed can be provided upon request.

5. Is the manuscript presented in an intelligible fashion and written in Standard English?

Reviewer 1:

Yes. Many thanks for your attention and time.

Reviewer 2:

Yes. Many thanks for your attention and time.

6. Review Comments to the Author

Reviewer 1: Yes.

We would like to thank the authors for addressing all comments. The manuscript now is more relevant and suitable for publication. Many thanks for your attention and time.

Reviewer 2: Yes.

As previously noted, it is recommended that the references in this section be updated to primarily include sources published within the last 3–5 years, in order to better reflect the current state of knowledge. Thanks for the comment from the respected referee. Older references have been updated. Currently, after updating the references, only references 29 and 31-41 are older, and given that these studies describe the initial design of the questionnaires used in this study in other contexts and their validity and reliability in Iran, it is not possible to update them.

---

## [Decision Letter · Decision Letter 3]

1 Feb 2026

The Influence of Social Support, Physical Activity, General Psychological Distress, and Demographic Characteristics on Self-Reported Health Status among Women in Iran

PONE-D-25-48364R3

Dear Dr. Darvishigilan,

We’re pleased to inform you that your manuscript has been judged scientifically suitable for publication and will be formally accepted for publication once it meets all outstanding technical requirements.

Kind regards,

Zahra Lorigooini

Academic Editor

PLOS One

Additional Editor Comments (optional):

Reviewers' comments:

Reviewer's Responses to Questions

**Comments to the Author**

Reviewer #2: All comments have been addressed

Reviewer #3: All comments have been addressed

2. Is the manuscript technically sound, and do the data support the conclusions?

Reviewer #2: Yes

Reviewer #3: Partly

3. Has the statistical analysis been performed appropriately and rigorously?

Reviewer #2: Yes

Reviewer #3: I Don't Know

4. Have the authors made all data underlying the findings in their manuscript fully available?

Reviewer #2: Yes

Reviewer #3: Yes

5. Is the manuscript presented in an intelligible fashion and written in standard English?

Reviewer #2: Yes

Reviewer #3: Yes

Reviewer #2: All comments have been addressed. We have no other suggestions for the author to address.

The manuscript now is suitable for publication.

Reviewer #3: The authors have satisfactorily addressed all previous comments. The manuscript is clear, compliant with PLOS policies, and suitable for publication. I recommend acceptance.

**Do you want your identity to be public for this peer review?** For information about this choice, including consent withdrawal, please see our Privacy Policy

Reviewer #2: **Yes:** Maha Dardouri

Reviewer #3: **Yes:** Mobin Ebrahimian

---

## [Editor Report · Acceptance letter]

PONE-D-25-48364R3

PLOS One

Dear Dr. Darvishigilan,

I'm pleased to inform you that your manuscript has been deemed suitable for publication in PLOS One. Congratulations! Your manuscript is now being handed over to our production team.

Kind regards,

on behalf of

Prof. Zahra Lorigooini

Academic Editor

PLOS One